# New Janus Tricyclic Laddersiloxanes: Synthesis, Characterization, and Reactivity

**DOI:** 10.3390/molecules28155699

**Published:** 2023-07-27

**Authors:** Yujia Liu, Midori Tokuda, Nobuhiro Takeda, Armelle Ouali, Masafumi Unno

**Affiliations:** 1Department of Chemistry and Chemical Biology, Gunma University, 1-5-1 Tenjin-cho, Kiryu 376-8515, Japan; midori.tokuda@gmail.com (M.T.); ntakeda@gunma-u.ac.jp (N.T.); 2ICGM, Univ. Montpellier, CNRS, ENSCM, 1919 Route de Mende, 34293 Montpellier, CEDEX 05, France

**Keywords:** tricyclic laddersiloxanes, Janus molecules, hydrosilylation, *syn*-type conformation, well-defined structures

## Abstract

The synthesis of four novel *syn*-type tricyclic laddersiloxanes bearing eight or six alkenyl groups is presented. These compounds possess reactive alkenyl groups on both the bridged and side silicon atoms, and their structures were determined through characterization using multinuclear 1D and 2D NMR spectroscopy, mass spectrometry, and elemental analysis techniques. To investigate their reactivity, the compounds were subjected to hydrosilylation using two different silanes, and the resulting fully hydrosilylated compounds were thoroughly analyzed. Remarkably, all the synthesized laddersiloxanes displayed high thermal stability, suggesting their potential as promising precursors for the development of new hybrid materials. Additionally, preliminary findings indicate the possibility of exploiting the reactivity difference between the alkenyl groups attached to the D- and T-unit silicon atoms for the synthesis of Janus molecules. These findings highlight the potential of the reported compounds as valuable building blocks in the construction of innovative materials.

## 1. Introduction

Silsesquioxanes are organosilicon compounds primarily consisting of silicon atoms bonded to three oxygen atoms and one organic substituent. Well-defined silsesquioxanes feature rigid, thermally stable, and chemically inert inorganic siloxane cores, with reactive and easily modifiable organic fragments attached to them [1,2]. These compounds exhibit unique hybrid properties, blending organic and inorganic characteristics, which make them promising building blocks for developing hybrid materials for various applications [3,4,5,6,7,8,9]. Among these compounds, laddersiloxanes, which are well-defined ladder-type silsesquioxanes, exhibit highly ordered double-chain structures and have the potential to form polymers [10,11]. Additionally, they possess the highest refractive indices among linear siloxanes, cyclic siloxanes, and cage silsesquioxanes [12]. Due to these exceptional properties, laddersiloxanes have attracted increasing attention over the past decade. The exploration of laddersiloxanes dates back to the 1960s when Brown first proposed an example of this structure [13]. Over the years, improved synthetic routes and characterization methods have been established, leading to the discovery of laddersiloxanes with up to 9 fused rings [10]. Among them, the *syn* or *anti*-type tricyclic laddersiloxanes with 6-8-6 or 8-8-8-membered fused rings have been the most intensively investigated (Figure 1, I.). The synthesis of 8-8-8 laddersiloxanes typically involves cyclotetrasiloxanes and dichloro- or dihydrosiloxanes [14,15,16,17,18,19]. More recently, these compounds have also been directly generated from silylated cyclotetrasiloxanes in the presence of HCl or a borane catalyst [20,21]. Notably, this method enabled the synthesis of a unique 12-8-12 tricyclic laddersiloxane, starting from an extended cyclotetrasiloxane [22]. Another approach to preparing 8-8-8 tricyclic laddersiloxanes is the oxidation of the corresponding tricyclic ladder oligosilanes [23]. For the synthesis of 6-8-6 tricyclic laddersiloxanes, cyclotetrasiloxanes, and dichloro- or dihydroxysilanes can be employed [19,24,25,26,27]. Of particular interest were *syn*-type tricyclic laddersiloxanes, which exhibited a structure similar to Janus molecules (Figure 1, III.). Similar to the Roman god Janus, these molecules possess two distinct faces and have garnered significant attention in scientific research and various fields due to their dissymmetric nature [28,29,30,31].

In 2019, the first synthesis and characterization of well-defined *syn*-type 6-8-6 tricyclic laddersiloxanes bearing four peripheral reactive substituents (vinyl or allyl groups) were reported [25]. The introduction of four chloro and four thiol substituents was successfully achieved through organic reactions, such as hydrosilylation and thiol-ene reactions [25,32]. Later, the synthesis of less symmetrical laddersiloxanes with only one vinyl group on each side silicon atom was described, and these compounds were found to be suitable as monomers for polymer preparation [26,33]. The resulting polymers exhibited unconventional conjugation behavior and demonstrated potential for use in semiconducting materials. Furthermore, functional groups were introduced onto the bridged silicon atoms, thereby expanding the range of possibilities [27]. However, despite these advancements, the exploration of *syn*-type tricyclic laddersiloxanes as Janus molecules remains relatively unexplored. Previous studies have reported *syn*-type tricyclic laddersiloxanes with reactive groups on either side of the bridged silicon atoms, although compounds featuring reactive groups on both the side and bridged silicon atoms have yet to be reported (Figure 1, I.).

In this work, we describe the synthesis of novel *syn*-type 6-8-6 tricyclic laddersiloxanes with reactive alkenyl groups on both the bridged and side silicon atoms (Figure 1, II.). Additionally, the eight or six alkenyl groups on these laddersiloxanes can be fully functionalized through hydrosilylation with silanes containing reactive substituents, thus, demonstrating their potential as precursors for the preparation of new hybrid materials. Moreover, a selective hydrosilylation trial was conducted on Janus laddersiloxanes that contained four vinyl groups on the bridged silicon atoms and four allyl groups on the side silicon atoms. The promising results obtained highlight the potential of this compound in the development of new Janus materials.

## 2. Results and Discussion

### 2.1. Preparation of Janus Tricyclic Laddersiloxanes Bearing Eight or Six Alkenyl Groups

The targeted *syn*-type tricyclic laddersiloxanes were synthesized from all-*cis*-tetravinylcyclotetrasiloxanolate [ViSi(OK)O]_4_ (**1**) [27]. A mixture of the freshly prepared and pre-dried [ViSi(OK)O]_4_ (**1**), distilled triethylamine, and anhydrous THF was prepared and cooled to 0 °C. Next, a solution of the corresponding dichlorosilane (specifically, dichlorodivinylsilane (**2**), diallyldichlorosilane (**3**), dichloromethylvinylsilane (**4**), or dichlorophenylvinylsilane (**5**)) in anhydrous THF was added, dropwise, at 0 °C, under an argon atmosphere. After the addition, the reaction mixture was stirred at room temperature for one hour. Then, the crude product obtained after extraction was purified using gel permeation chromatography (GPC), resulting in the isolation of pure target compounds **6**, **7**, **8**, and **9**, with yields ranging from 27 to 53% (Figure 2). Importantly, the reaction conditions employed preserved the all-*cis* configuration of the cyclotetrasiloxanolate, and all the synthesized compounds (**6**–**9**) exhibited a *syn*-type conformation.

The ^29^Si NMR spectrum displayed two distinct peaks at −69.15 ppm and −35.21 ppm (Figure 1a), corresponding to the bridged (T-unit) silicon atoms (Si in pink) and the side (D-unit) silicon atoms (Si in blue), respectively, which is consistent with previously reported analogous laddersiloxanes [25,27]. Analysis of laddersiloxane **6** using ^1^H NMR revealed a range of signals from 5.95 to 6.16 ppm, corresponding to the vinyl groups (see Appendix A). The ^13^C NMR spectrum demonstrated magnetic inequivalence for the two vinyl substituents connected to the same D-unit silicon atom (see Appendix A), which is in accordance with previous reports on this type of structure [25]. There were four distinct signals for the carbon atoms of the peripheral vinyl groups, resulting in a total of six carbon signals for compound **6**.

Moving on to laddersiloxane **7**, the ^29^Si NMR spectrum showed two single peaks at −69.17 ppm (T-unit silicon) and −15.78 ppm (D-unit silicon) (Figure 1b), which matched well with the values previously reported for analogous laddersiloxanes [25,27]. The ^1^H NMR spectrum (see Appendix A) clearly distinguished signals assigned to the vinyl and allyl groups. Specifically, the multiplets at 4.93–5.04 ppm and 5.75–5.84 ppm represented the allyl groups connected to the D-unit silicon atoms, while the three clear doublet–doublet (dd) patterns at 5.95, 6.07, and 6.14 ppm corresponded to the four chemically equivalent vinyl groups on the T-unit silicon atoms (confirmed by the ^1^H-^1^H COSY and ^1^H-^13^C HSQC NMR spectra; see Appendix A). Additionally, two distinctive doublet signals at 1.74 ppm and 1.80 ppm indicated the presence of two types of methylene groups adjacent to the D-unit silicon. The ^13^C NMR spectrum (see Appendix A) exhibited two signals for the carbon atoms in the methylene groups and six signals for the carbon atoms in the olefinic moieties, confirming the *syn*-type conformation of tricyclic laddersiloxane **7**.

Compounds **8** and **9** possessed two different substituents on the D-unit silicon atoms, which resulted in less symmetrical structures. Each compound had three possible stereoisomers [26]. The ^29^Si NMR spectrum of compound **8** clearly revealed four singlets for the D-unit silicon atoms at −19.29, −19.39, −19.64, and −19.77 ppm, as well as T-unit silicon atoms ranging from −69.19 to −69.25 ppm, thereby indicating the presence of three isomers (Figure 1c). This was further supported by the ^1^H and ^13^C NMR analyses (see Appendix A), which displayed four signals for the methyl groups. The formation of three isomers of compound **9** was also observed in the corresponding ^29^Si NMR spectrum, which showed four singlets for both the D-unit (−33.86, −33.96, −34.02, −34.38 ppm) and T-unit (−68.40, −68.86, −69.00, −69.17 ppm) silicon atoms (Figure 1d).

To verify the structures of these four laddersiloxanes, matrix-assisted laser desorption/ionization coupled time-of-flight (MALDI-TOF) mass spectrometry and elemental analyses were performed, which yielded experimental results in good accordance with the calculated values (see Appendix A). These compounds were also analyzed by 2D NMR techniques, including ^1^H-^1^H COSY and ^1^H-^13^C HSQC, to further confirm their structures (see Appendix A). As an example, a predicted representative crystal structure of compound **7** was illustrated in Figure 2a.

### 2.2. Full Functionalization of Janus Tricyclic Laddersiloxanes Bearing Eight or Six Alkenyl Groups

*Syn*-type tricyclic laddersiloxanes (**6**–**9**) possess either eight or six reactive double bonds, making them potentially valuable precursors for creating new hybrid materials. To assess the reactivity of the surrounding alkenyl substituents, compounds **6**, **7**, and **8** were hydrosilylated with dimethylphenylsilane (65 °C for 18 h) (Figure 3). ^1^H NMR analysis of compounds **7** and **8** confirmed that the reaction occurred quantitatively, as evidenced by the disappearance of signals corresponding to the alkenyl groups. In the case of compound **6**, an increase in temperature from 65 to 100 °C was required to complete the transformation of the vinyl groups. After purification using GPC, pure functionalized products **10** and **11** were successfully obtained (82% and 40%, respectively), while compound **12** was obtained quantitatively (99%), as a mixture of three stereoisomers without GPC purification.

The ^29^Si NMR spectrum of compound **10** revealed three groups of signals representing the silicon atoms of the T-unit (Si in pink), D-unit (Si in blue), and carbosilane moieties (Si in green) (Figure 3a). The chemical shifts of the T- and D-unit silicon atoms in compound **10** were downfield shifted to −55.06 and −7.94 ppm, respectively, compared to −69.15 and −35.21 ppm, respectively, in compound **6**, due to the reduction in vinyl groups. The appearance of new signals at −1.12, −1.18, and −1.31 ppm in the ^29^Si NMR spectrum indicated the presence of silicon atoms in the 3 types of carbosilane moieties (Figure 3a). Similarly, in compound **11**, a downfield shift was observed in both the chemical shifts of the T- and D-unit silicon atoms (−55.41 and −8.46 ppm) compared to −69.17 and −15.78 ppm in compound **7**, indicating the successful reduction of all olefins (Figure 3b). The ^29^Si NMR spectrum of compound **11** showed 3 singlets, which corresponded to the silicon atoms of the carbosilane parts: 1 (Si in green) at −1.16 ppm, connected to the T-unit silicon atoms, and 2 (Si in red) at −3.80 and −3.91 ppm, attached to the D-unit silicon atoms (Figure 3b). In the case of compound **12**, the ^29^Si NMR spectrum exhibited 4 distinct singlets at −54.95, −54.99, −55.31, and −55.39 ppm, indicating the presence of 3 isomers, with 2 patterns of signals around −1.1 and −6.5 ppm, assigned to the carbosilane and D-unit silicon atoms, respectively (Figure 3c). The ^1^H and ^13^C NMR spectra of compounds **10**, **11**, and **12** exhibited signals corresponding to methylene groups resulting from the hydrosilylation of the olefinic substituents (see Appendix A).

Compounds **7** and **8**, which were more easily hydrosilylated, were selected for hydrosilylation with chloromethyl(dimethyl) silane, which contains a reactive chloro group. The reaction conditions were similar to those used for dimethylphenylsilane, although at a lower temperature (40 °C) (Figure 4). After 18 h, the reactions with both compounds **7** and **8** were complete and all olefinic groups were fully consumed. The ^29^Si NMR spectrum of compound **13** displayed downfield-shifted peaks for the T- and D-unit silicon atoms (−55.58 and −8.58 ppm, respectively), compared to compound **7**. Additionally, 3 peaks at 5.38 ppm were assigned to the silicon atoms of the carbosilane on the bridged silicon atoms, and 2.99 and 2.93 ppm were assigned to the silicon atoms of the carbosilane on the side silicon atoms (Figure 3d). ^1^H NMR analysis of compound **13** (see Appendix A) revealed 3 singlets at 2.79, 2.774, and 2.770 ppm, which corresponded to the 3 types of protons adjacent to the chloro groups. The signals of the methylene groups between the two silicon atoms, resulting from the reduction in olefins, appeared in the range of 0.63-0.75 ppm and 1.44–1.56 ppm. The hydrosilylated product (**14**) from compound **8** was formed from a mixture of 3 stereoisomers, which was supported by the multinuclear NMR results. For instance, in the ^29^Si NMR spectrum of **14**, 3 distinct singlets were observed at −55.12, −55.22, and −55.58 ppm for the T-unit silicon atoms, and at −6.17, −6.57, and −6.92 ppm for the D-unit silicon atoms (Figure 3e). It should be noted that theoretically, four singlets were expected, yet only two of them appeared to overlap in the experimental spectrum. The functionalized laddersiloxanes (**10**–**14**) obtained through hydrosilylation were also characterized using MALDI-TOF mass spectrometry and elemental analyses. The experimental results were consistent with the theoretically calculated results (see Appendix A). These compounds were also analyzed by 2D NMR techniques, including ^1^H-^1^H COSY and ^1^H-^13^C HSQC, to further confirm their structures (see Appendix A). As an example, a predicted representative crystal structure of compound **11** is illustrated in Figure 2b.

### 2.3. Thermal Properties of Laddersiloxanes ***6**–**14***

To investigate the thermal properties of the synthesized laddersiloxanes (**6**–**14**), thermogravimetric analysis (TGA) was performed under a nitrogen flow (250 mL min^−1^) at a rate of 10 °C min^−1^. Figure 4 and Figure 5 illustrate the TGA results for each laddersiloxane.

Laddersiloxanes (**6**–**9**) containing either eight or six alkenyl groups exhibited a two-step weight loss profile (Figure 4). The first step corresponds to the cleavage of the C–Si bond, while the second step could be attributed to the cleavage of the side Si–O bond. Compounds **7** and **9** showed similar initial decomposition temperatures, both higher than compound **6**, which in turn was higher than compound **8**. Furthermore, the decomposition temperatures at 5% weight loss of the initial mass (T_d5_) for compounds **7** and **9** were higher (196 °C and 198 °C, respectively) than for compound **6** (172 °C), while compound **8** exhibited the lowest T_d5_ value (148 °C). Of particular interest was the residue weight of compound **6** at 1000 °C, which amounted to 80% of the initial weight, surpassing the weight percentage of the siloxane core (Si + O: 58%) by approximately 20%. This significant value suggests that compound **6** may undergo self-polymerization at high temperatures due to the presence of two adjacent vinyl groups on each side of the silicon atom, thereby leading to the formation of highly thermally stable polymers.

The thermal stability of the hydrosilylated laddersiloxanes (**10**–**14**) was also explored using TGA. These compounds showed a one-step weight loss profile (Figure 5). The order of the initial decomposition temperatures was as follows: compound **10** ≈ compound **11** > compound **12** > compound **13** > compound **14**. Compound **11** exhibited the highest T_d5_ value (422 °C) among all the synthesized laddersiloxanes. Compounds **10** and **12** displayed comparable T_d5_ values at 340 °C and 348 °C, respectively. In the absence of phenyl groups, compounds **13** and **14** exhibited slightly lower thermal stabilities, with T_d5_ values of 333 and 323 °C, respectively. A similar effect of phenyl groups on thermal stability has already been observed [26].

### 2.4. Trials of Selective Functionalization of Janus Tricyclic Laddersiloxane ***7***

All the synthesized *syn*-type tricyclic laddersiloxanes exhibited distinct up and down faces when considering the plan of the central cyclotetrasiloxane core (Figure 1). Of particular interest was compound **7**, which possessed two types of reactive substituents: vinyl groups on the upper face and allyl groups on the bottom face. Although vinyl and allyl substituents share the same functional group, namely a double bond, their reactivity towards organic transformations may vary.

To demonstrate this concept, we conducted a hydrosilylation reaction using compound **7** and dimethylphenylsilane (in an insufficient amount) under similar reaction conditions as described above. ^1^H NMR analysis of the resulting crude product (**15**) revealed the complete disappearance of signals corresponding to the vinyl groups in the range of 5.95–6.14 ppm, while the peaks corresponding to the allyl groups remained partially observable (Figure 6). In the ^29^Si NMR spectrum for the crude product **15** (see Appendix A), the peak at −69.17 ppm, assigned to the T-unit silicon atoms connected to the vinyl groups, in compound **7** disappeared, while new peaks appeared around −55.2 ppm and −1.1 ppm, corresponding to the T-unit and carbosilane silicon atoms, resulting from the hydrosilylation of the vinyl groups. Additionally, the signals of the D-unit silicon atoms (−15.78 ppm) connected to the allyl groups of compound **7** shifted to approximately −8.4 ppm, −12.9 ppm, and −17.2 ppm, and new signals were observed around −3.8 ppm. These changes in the ^29^Si NMR chemical shifts indicate that after hydrosilylation with six equivalents of dimethylphenylsilane, all four vinyl groups on the bridged silicon atoms were completely substituted, while the four allyl groups on the side silicon atoms were only partially substituted, resulting in different substituted products (Figure 7). This result suggests that the reactivity of the vinyl groups on the bridged silicon atoms in the hydrosilylation reaction is higher than the allyl groups on the side silicon atoms. Furthermore, MALDI-TOF mass spectrometry was performed on crude product **15** (see Appendix A), and supported the NMR results. Mass spectrometry analysis revealed that the main products were compounds **15b** and **15c**, which correspond to the mono- and di-substituted allyl groups, respectively. Theoretically, compound **15b** has 2 stereoisomers, whereas **15c** has 4 stereoisomers. The presence of multiple signals for each type of silicon atom suggested the formation of stereoisomers for each compound. Additionally, minor products, such as compounds **15a** and **15d**, which contained 4 and 3 allyl groups, respectively, were also detectable. Although the isolation of different compounds from crude product **15** has not been tested, these results indicate that the functionalization of compound **7** is tunable and that this compound holds promise as a precursor in the preparation of new Janus materials.

## 3. Materials and Methods

### 3.1. General Considerations

All reactions were performed under an argon atmosphere using the standard Schlenk technique unless stated otherwise. THF and toluene were dried using an mBRAUN purification system. Triethylamine was distilled from potassium hydroxide and stored in potassium hydroxide under an argon atmosphere with protection from light. Triethoxyvinylsilane was purchased from Asahikasei Wacker Silicon Co., Ltd. (Tokyo, Japan); dichlorodivinylsilane was purchased from Shin-Etsu Chemical Co., Ltd. (Tokyo, Japan); diallyldichlorosilane and dichlorophenylvinylsilane were purchased from Gelest, Inc. (Morrisville, PA, USA); dichloromethylvinylsilane, dimethylphenylsilane, and chloromethyl(dimethyl)silane were purchased from TCI Co., Ltd. (Tokyo, Japan); Karstedt’s catalyst (in xylene, 2% Pt) was purchased from Sigma-Aldrich (St. Louis, MO, USA). All reagents were used as received without further purification.

Fourier transform nuclear magnetic resonance (NMR) spectra were obtained using a JEOL JNM-ECA 600 (^1^H at 600.17 MHz, ^13^C at 150.91 MHz, ^29^Si at 119.24 MHz) NMR instrument. For ^1^H NMR, chemical shifts were reported as δ units (ppm) relative to SiMe_4_ (TMS), and the residual solvent peaks were used as standards. For ^13^C NMR and ^29^Si NMR, chemical shifts were reported as δ units (ppm) relative to SiMe_4_ (TMS). The residual solvent peaks were used as standards, and the spectra were acquired with complete proton decoupling. Matrix-assisted laser desorption/ionization coupled time-of-flight (MALDI-TOF) mass analyses were performed by a Shimadzu (Kyoto, Japan) AXIMA Performance instrument, using 2,5-dihydroxybenzoic acid (dithranol) as the matrix and AgNO_3_ as the ion source. All used reagents were of analytical grade. Elemental analyses were performed at the Center for Material Research by Instrumental Analysis (CIA), Gunma University, Japan. Infrared (IR) spectra were measured using a Shimadzu IRSpirit FTIR spectrometer. TGA was performed using a Rigaku (Tokyo, Japan) thermogravimetric analyzer (Thermoplus TG-8120). The investigations were carried out under nitrogen flow (250 mL min^−1^) or airflow (300 mL min^−1^) at a heating rate of 10 °C min^−1^. All samples were measured in a temperature range of 50 to 1000 °C, with a 5 min hold at 1000 °C. The weight loss and heating rate were continuously recorded during the experiment. Gel permeation chromatography (GPC) was performed using a Japan Analytical Industry (Tokyo, Japan) LaboACE LC-5060 instrument.

### 3.2. Synthetic Procedures of Compounds ***6**–**15***

Typical protocol for the synthesis of 6-8-6 tricyclic laddersiloxanes (**6**–**9**)

An argon-purged 3-necked flask equipped with a magnetic stirring bar and an addition funnel was charged with all-*cis*-[ViSi(OK)O]_4_ (0.5 g, 1.0 mmol), anhydrous THF (20 mL), and distilled Et_3_N (0.42 mL, 3.0 mmol). A solution of dichlorosilane (3.0 mmol) in anhydrous THF (45 mL) was prepared and added dropwise, via the additional funnel to the flask at 0 °C, under an argon atmosphere. After the addition, the reaction mixture was stirred at 25 °C for 1 h. Then, a saturated NH_4_Cl aqueous solution was added to the reaction medium to neutralize Et_3_N, and chloroform was added to extract the product three times. Then, the gathered organic layer was washed with brine three times, dried over anhydrous Na_2_SO_4_, and concentrated on a rotary evaporator to afford the crude product as a slightly yellow viscous oil, which was purified by GPC (CHCl_3_) to provide the corresponding pure product (laddersiloxane **6**: colorless solid, 0.15 g, 30%; laddersiloxane **7**: colorless liquid, 0.30 g, 53%; laddersiloxane **8**: colorless solid, 0.20 g, 41%; laddersiloxane **9**: colorless solid, 0.166 g, 27%).

Synthesis of 6-8-6 tricyclic laddersiloxane (**10**)

An argon-purged Schlenk flask equipped with a stir bar was charged with 6 (0.026 g, 0.05 mmol), anhydrous toluene (0.3 mL), and dimethylphenylsilane (92 µL, 0.6 mmol). Karstedt’s catalyst (2% Pt, commercial bottle, diluted 100 times in anhydrous toluene under argon, 115 μL, 0.1 μmol Pt) was added to the mixture under an argon atmosphere at 25 °C. After the addition, the mixture was heated to 100 °C and stirred at 100 °C for 20 h. After the reaction, the mixture was cooled to room temperature and passed through a silica plug, which was washed with toluene and dichloromethane. The solvents were removed using a rotary evaporator to afford the crude product as a viscous colorless oil (87 mg). The crude product was purified using GPC (eluent: CHCl_3_), and pure product 10 was obtained as a colorless liquid (66 mg, 82%).

Synthesis of 6-8-6 tricyclic laddersiloxane (**11**)

An argon-purged Schlenk flask equipped with a stir bar was charged with **7** (0.028 g, 0.050 mmol), anhydrous toluene (0.30 mL), and dimethylphenylsilane (92 µL, 0.60 mmol). Karstedt’s catalyst (2% Pt, commercial bottle, diluted 100 times in anhydrous toluene under argon, 115 μL, 0.1 μmol Pt) was added to the mixture under an argon atmosphere at 25 °C. After the addition, the mixture was heated to 65 °C and stirred at 65 °C for 18 h. After the reaction, the mixture was cooled to room temperature and passed through a silica plug, which was washed with toluene and dichloromethane. The solvents were removed using a rotary evaporator to obtain viscous brown oil (90 mg). The crude product was purified using GPC (CHCl_3_), and pure product **11** was obtained as a colorless liquid (32 mg, 40%).

Synthesis of 6-8-6 tricyclic laddersiloxane (**12**)

An argon-purged Schlenk flask equipped with a stir bar was charged with 8 (0.024 g, 0.05 mmol), anhydrous toluene (0.3 mL), and dimethylphenylsilane (61 µL, 0.45 mmol). Karstedt’s catalyst (2% Pt, commercial bottle, diluted 100 times in anhydrous toluene under argon, 57 μL, 0.05 μmol Pt) was added to the mixture under an argon atmosphere at 25 °C. After the addition, the mixture was heated to 65 °C and stirred at 65 °C for 18 h. After the reaction, the mixture was cooled to room temperature and passed through a silica plug, which was washed with toluene and dichloromethane. The solvents were removed using a rotary evaporator to obtain pure product 12 as a viscous yellow oil (64 mg, 99%).

Synthesis of 6-8-6 tricyclic laddersiloxane (**13**)

An argon-purged Schlenk flask equipped with a stir bar was charged with **7** (0.028 g, 0.050 mmol), anhydrous toluene (0.30 mL), and chloromethyl(dimethyl)silane (73 µL, 0.60 mmol). Karstedt’s catalyst (2% Pt, commercial bottle, diluted 100 times in anhydrous toluene under argon, 115 μL, 0.1 μmol Pt) was added to the mixture under an argon atmosphere at 25 °C. After the addition, the mixture was heated to 40 °C and stirred at 40 °C for 18 h. After the reaction, the mixture was cooled to room temperature and passed through a silica plug, which was washed with toluene and dichloromethane. The solvents were removed using a rotary evaporator to obtain **13** as a colorless solid (42 mg, 60%).

Synthesis of 6-8-6 tricyclic laddersiloxane (**14**)

An argon-purged Schlenk flask equipped with a stir bar was charged with 8 (0.024 g, 0.05 mmol), anhydrous toluene (0.3 mL), and chloromethyl(dimethyl)silane (55 µL, 0.45 mmol). Karstedt’s catalyst (2% Pt, commercial bottle, diluted 100 times in anhydrous toluene under argon, 57 μL, 0.05 μmol Pt) was added to the mixture under an argon atmosphere at 25 °C. After the addition, the mixture was heated to 40 °C and stirred at 40 °C for 18 h. After the reaction, the mixture was cooled to room temperature and passed through a silica plug, which was washed with toluene and dichloromethane. The solvents were removed using a rotary evaporator to obtain pure product 14 as a viscous colorless oil (57 mg, 99%).

Synthesis of 6-8-6 tricyclic laddersiloxane (**15**)

An argon-purged Schlenk flask equipped with a stir bar was charged with **7** (0.056 g, 0.1 mmol), anhydrous toluene (0.3 mL), and dimethylphenylsilane (92 µL, 0.6 mmol). Karstedt’s catalyst (2% Pt, commercial bottle, diluted 100 times in anhydrous toluene under argon, 57 μL, 0.05 μmol Pt) was added to the mixture under an argon atmosphere at 25 °C. After the addition, the mixture was heated to 65 °C and stirred at 65 °C for 18 h. After the reaction, the mixture was cooled to room temperature and passed through a silica plug, which was washed with toluene and dichloromethane. The solvents were removed using a rotary evaporator to obtain crude product **15** as a viscous yellow oil (135 mg).

## 4. Conclusions

In conclusion, four novel *syn*-type Janus tricyclic laddersiloxanes (**6**–**9**), bearing either eight or six alkenyl groups, were successfully synthesized and comprehensively characterized. The structures of these compounds were determined using multinuclear 1D or 2D NMR spectroscopy, mass spectrometry, and elemental analysis techniques. Furthermore, hydrosilylation reactions of compounds **6**–**8** with 2 different silanes were carried out successfully, resulting in the formation of fully hydrosilylated compounds (**10**–**14**). Notably, all the synthesized laddersiloxanes exhibited high thermal stability, indicating their potential as promising precursors for the development of new hybrid materials. Moreover, preliminary results suggest the possibility of exploiting the reactivity difference between the alkenyl groups attached to the D- or T-unit silicon atoms for the development of Janus materials. This creates opportunities for further exploration and the utilization of compound **7** in the design and fabrication of innovative materials. Overall, this study represents the first synthesis of *syn*-type tricyclic laddersiloxanes with reactive groups on both bridged and side silicon atoms. The findings presented in this study contribute to the expanding knowledge in the field of laddersiloxanes and pave the way for future research and application opportunities in materials science and related disciplines.

## Data Availability

All data and material described in this work are available in this article or in Appendix A.

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
