# Peer review of "New Janus Tricyclic Laddersiloxanes: Synthesis, Characterization, and Reactivity"

_molecules, 2023, doi:10.3390/molecules28155699_

Round 1

Reviewer 1 Report

This is a very well prepared paper describing the synthesis of ladder siloxanes.  The style and experimental details are quite clear and the work will be of interest to those in the field.  The scope is a bit limited but it does not overstate the conclusions.  Further work on differential reactivity of the olefin linkages is likely to be forthcoming, but there should be no need to delay publication before these results.

While the analytical data is sufficient to fully characterize the products I think the inclusion of 2D NMR data particularly COSY and HSQC (proton and carbon) would enhance the paper.

Reviewer 2 Report

In this work Unno et al. reported the efficient synthesis of syn-type Janus tricyclic laddersiloxanes with eight or six terminal alkenyl groups and subsequent hydrosilylation reactions to generate fully hydrosilylated  compounds. All the materials have been well characterized and some solid spectrascopy evidence has been provided to illustrate the complex and diverse Si sheletons. It will be great if a representative crystal structure of these compounds could be provided. This research is of vital importance in consideration of the potential utilization of these laddersiloxanes as building blocks for novel hybrid materials. The manuscript is well written and organized. I would recommend its publication on Molucules.

Some minor editing of the English language is required.
